# Yaks Are Dependent on Gut Microbiota for Survival in the Environment of the Qinghai Tibet Plateau

**DOI:** 10.3390/microorganisms12061122

**Published:** 2024-05-31

**Authors:** Runze Wang, Binqiang Bai, Yayu Huang, Allan Degen, Jiandui Mi, Yanfeng Xue, Lizhuang Hao

**Affiliations:** 1Key Laboratory of Plateau Grazing Animal Nutrition and Feed Science of Qinghai Province, State Key Laboratory of Plateau Ecology and Agriculture, Qinghai University, Xining 810016, China; runzewang2000@foxmail.com (R.W.); binqiangbai@163.com (B.B.); 2PEGASE, INRAE, Institut Agro, 35590 Saint-Gilles, France; yayu.huang.fr@gmail.com; 3Desert Animal Adaptations and Husbandry, Wyler Department of Dryland Agriculture, Blaustein Institutes for Desert Research, Ben-Gurion University of the Negev, Beer Sheva 8410500, Israel; degen@bgu.ac.il; 4State Key Laboratory for Animal Disease Control and Prevention, College of Veterinary Medicine, Lanzhou University, Lanzhou 730000, China; mijiandui@gmail.com; 5College of Animal Science and Technology, Anhui Agricultural University, Hefei 230036, China; xueyanfeng1990@163.com

**Keywords:** yak, microbial interactions, Qinghai–Tibetan plateau

## Abstract

The yak (*Poephagus grunniens*) has evolved unique adaptations to survive the harsh environment of the Qinghai–Tibetan Plateau, while their gut microorganisms play a crucial role in maintaining the health of the animal. Gut microbes spread through the animal population not only by horizontal transmission but also vertically, which enhances microbial stability and inheritance between generations of the population. Homogenization of gut microbes in different animal species occurs in the same habitat, promoting interspecies coexistence. Using the yak as a model animal, this paper discusses the adaptive strategies under extreme environments, and how the gut microbes of the yak circulate throughout the Tibetan Plateau system, which not only affects other plateau animals such as plateau pikas, but can also have a profound impact on the health of people. By examining the relationships between yaks and their gut microbiota, this review offers new insights into the adaptation of yaks and their ecological niche on the Qinghai–Tibetan plateau.

## 1. Introduction

Yaks (*Poephagus grunniens*) diverged from cattle between one and five million years ago, and it has been suggested that yaks are more closely related to bison than to other Bovidae [1]. As an ancient domestic animal of the Tibetan Plateau, the yak has inhabited this land since ancient times, being domesticated by the Qiang people 10,000 years ago [2]. They are crucial for the livelihoods of the inhabitants, providing meat, milk, dung and wool [3]. The severe habitat, namely extreme cold, a low oxygen content, strong ultraviolet light and a short growing season, has shaped the yak’s adaptation to the harsh environment.

The gut microorganisms, also known as ‘the second genome’, are considered to be the ‘second digestive organ’ of ruminants, due to the close association between them and the host [4]. Gut microorganisms influence host adaptation, and help the host cope with challenges such as food shortage, phenotypic plasticity, and adaptive immunity [5]. Yaks are an essential component of the plateau ecosystem and interact with organisms, as well as with the environment through microorganisms, which play a central role in the exchange of matter and energy, and transfer information between ecosystems and even between animals. The unique adaptive strategies evolved by yaks cannot be separated from their gut microbes [6], in particular, the seasonal variation in their gut microbial composition with changes in forage availability [7]. Yaks have low nitrogen and carbon requirements under microbial mediation [8], and the gut microorganisms influence the quality of yak products [9]. Studies thus far have focused on a single aspect of yak–microbe interactions, while a systematic review of yak microbes is still lacking. This paper aims to fill this important gap by presenting a comprehensive review on the co-evolution of yaks and microbes, the microbial mechanisms behind yak response to pasture scarcity, the impact of yak microorganisms on livestock products, and the relationship between yak health and microbes.

## 2. Synergistic Evolution between Yaks and Microorganisms

Throughout their long history, yaks co-evolved with microorganisms, which enabled them to adapt to the harsh environment. This co-evolution can be witnessed through the current yak–microbe relationships, where gut microorganisms colonize the yak from birth [10]. During the early stages of rumen development, yaks exhibit functional links and synergistic effects between rumen epithelial cell genes and microbial genes [11]. Compared to adult yaks, pathways related to transport and catabolism are up-regulated in the rumen microbes of 5- to 180-day-old calves, which is beneficial to calf health and improves calf survival [12]. Yak rumen microorganisms alter accordingly to cope with changing diets at different growth stages, with greater relative abundances of Ascomycetes and Mesomycetes, that is, fungi adapted to liquid rations during juvenile years [13,14]. The fungal structure changes continually; at the grazing stage, after 90 days of age, there is an increase in the relative abundance of *Thelebolus*, a genus that contributes to host immunity and by 180 days of age, there is an increase in the relative abundance of *Penicillium*, a genus with a strong cellulose-degrading capacity [15]. In addition, yaks have adapted to the plateau environment by preferentially harboring protozoa such as rumen ciliates that have evolved to digest mainly fibers [16]. Yaks also alter rumen protozoa at 90 days of age as the dietary fiber content increases, with an increase in the abundance of *Dasytricha* spp, and a decrease in *Entodinium* [17], while *Isotricha* spp. has also been detected, enhancing the utilization of carbohydrates [18].

The co-evolution of yaks with microbiota has resulted in specific rumen microbial maturation strategies. Yak rumen archaea reach full maturity at approximately five years of age, and other microbiota mature between 5 and 8 years of age, corresponding to the peak growth period of yaks [18]. These insights into the co-evolutionary dynamics between yaks and their rumen microbiota not only highlight the intricate adaptations that support survival in extreme environments but also provide valuable perspectives for improving livestock management and conservation strategies.

## 3. Microbial Mechanisms of Yaks Coping with Pasture Scarcity

Seasonal pasture availability on the Qinghai–Tibetan plateau fluctuates greatly. During the long, cold winter, vegetation is sparse, with high fiber and low protein content, and yaks generally lose substantial body weight [7]. The warm season is the peak period for growth of pasture grasses, and yaks generally gain body weight. During the long evolutionary process, yak rumen microorganisms have gradually developed an adaptive mechanism by altering the composition according to the fluctuating pasture availability. During the summer, when pasture is abundant, the ability of yak rumen microbes to obtain energy from the diet increases substantially, which promotes compensatory growth in the renourishment process [19]. Conversely, during the cold season, when forage is scarce, yak rumen microbes enhance the ability to digest and utilize low-quality forage, thereby mitigating the effects of poor, fibrous forage [20]. Consequently, yaks cope with seasonal fluctuations in forage resources, at least in part, by their microbes.

### 3.1. Microbiological Mechanisms of Yaks with High Nutrient Digestibility

#### 3.1.1. Mechanisms of Yak Rumen Bacteria to Improve Nutrient Digestibility

The feeding modes of yaks on the Tibetan Plateau include grazing, housed feeding, and semi-housed feeding. However, regardless of the feeding mode, the dominant rumen bacterial flora of yaks do not change substantially, and at the phylum level, Firmicutes and Bacteroidetes dominate. Most Firmicutes play an essential role in plant cell wall degradation and the activity of Firmicutes increases the production rate of monosaccharides and VFAs, which aids in the absorption and utilization of nutrients [21,22,23]. The *Christensenellaceae_R7_group*, which enhances the degradation of cellulose, is the dominant genus of Firmicutes [1,24]. Bacteroidetes enhance the utilization of carbohydrates by degrading non-fiber complex polysaccharides chemosynthetically, maintain intestinal homeostasis, and are involved in protein hydrolysis [25,26,27]. Both Firmicutes and Bacteroidetes are closely related to the metabolism of fiber and non-fiber feed nutrients and enable yaks to better digest forage [28,29]. The dominant bacteria in yaks at the family level and their main functions are presented in Table 1.

#### 3.1.2. Mechanisms by Which Yak Rumen Fungi Improve Nutrient Digestibility

Ruminal fungi in yaks can efficiently degrade lignocellulose [30], in particular, the anaerobic fungi *Orpinomyces*, the dominant genus [31]. Bacteria and fungi in the yak rumen can have synergistic effects, and the presence of some fungi facilitates the bacterial digestion of cellulose [18]. For example, *Penicillium* is correlated positively with Firmicutes, which degrades mainly cellulose [32]. Fungi can also form associations with bacteria attached to food particles and extract energy from the bacteria [16]. Co-cultivation of methanogens with anaerobic fungi leads to changes in the metabolic pathways of the fungi, with a better ability to degrade lignocellulose and produce methane and acetic acid than anaerobic fungi alone [30].

#### 3.1.3. Rumen Microbial Secretion of Enzymes to Improve Nutrient Digestibility

Enzymes encoded by the yak rumen microbiome, similar to how microbial genes in yaks guide enzyme formation, play a crucial role in the digestion of feed [25]. Compared with cattle, the rumen microbiota of yaks encode cellulase, hemicellulase and polysaccharide lyase (PL family) with greater abundances, which contributes to cellulose degradation. However, the rumen microbiome encode fewer glucoside hydrolase glycolytases (GH family) with starch digestion ability and carbohydrate-binding modules (CBM family) with the ability to improve the catalytic efficiency of carbohydrate enzymes [33,34]. As a result, the degradation of starch in the rumen decelerates, allowing more starch to reach the small intestine, which uses starch more efficiently than the rumen [35]. The rumen fungus *Neocallimastix* sp. YAK11, isolated from feces, has high feruloyl esterase, acetyl esterase, and xylanase activities, which enable yaks to utilize low-quality roughage to a greater extent [20].

### 3.2. Microbiological Strategies of Yaks to Cope with Seasonal Variation in Forage

Yak gut microbiota adapt to transitions between diets with nutritional differences in about 16 days, which enables yaks to rapidly adapt to a wide range of forage sources and seasonal pasture fluctuations on the Qinghai–Tibetan plateau. Although the structure of the yak gut microbial community undergoes substantial changes during the transition process, it stabilizes over time [36]. Therefore, although yaks experience large fluctuations in body condition, they can successfully adapt to the harsh environment of the Qinghai–Tibetan plateau in the absence of supplemental feeding through alterations in the structure of their gut microbial communities, as well as seasonal variations in gut shape.

#### 3.2.1. Seasonal Gut Patterns in Yaks

The proportion of ruminal Proteobacteria in yaks does not change throughout their entire growth stage, while members of the Proteobacteria class exhibit metabolic flexibility. Therefore, they can adapt to changes in substrates and energy sources by altering their gene pool, allowing yaks to adapt to dietary changes [37]. To adapt to the seasonal pasture fluctuations, yaks have evolved seasonal gut dynamics: gut enterotype 1, mainly during the cold season, is dominated by Akkermansia, which contributes to efficient nitrogen utilization, and by uncultured eubacterium WCHB1-41; gut enterotype 2, mainly in the warm season, is dominated by Ruminococcaceae_UCG-005, which is very efficient with low-protein, low-fiber diets [7].

#### 3.2.2. Dominant Flora Established by Yaks in Response to the Harsh Environment

Yaks maintain a stable microbial ecosystem in their rumen to adapt to dietary shifts across different grazing seasons, with microbial populations varying in response to these changes [38]. For instance, there is a greater prevalence of Verrucomicrobia in the cold than the warm season in yak rumen. Verrucomicrobia secretes glycosidic hydrolases that play a key role in the degradation of carbohydrates [39]. The predominant species within this phylum, *Akkermansia muciniphila*, enlarges the surface area for intestinal absorption, aids in regulating the intestinal barrier, and modulates the microbial community, thereby improving energy utilization [40]. Consequently, the increased presence of Verrucomicrobia during the winter aids yaks in managing energy more effectively under cold conditions. Furthermore, the proportion of certain methanogenic bacteria, such as those from Euryarchaeota and Methanobrevibacter, declines in cold seasons [41], which helps minimize energy losses during severe winter conditions [42].

Yak rumen microbes are influenced by dietary factors. For example, the greater organic matter content in a high-fiber diet enhances the growth of fibrolytic bacteria, and a high-energy diet rich in starch and fat increases the abundance of Prevotellaceae [43,44]. Moreover, the greater abundance of Clostridial, Ruminococcaceae and Prevotella in the intestinal tract of yaks than cattle resulted in greater concentrations of short-chain fatty acids (SCFAs) in the rumen of yaks than in cattle [45]. The SCFAs provide close to 70% of the energy requirements of ruminants [46]. The microbial mechanisms by which yaks cope with pasture scarcity are summarized in Figure 1.

## 4. Microbiological Mechanisms of Yak Health Maintenance

Yaks live in the harsh environment of the Qinghai–Tibetan plateau, where many factors, such as stress and sudden weather changes, can threaten their health. Yaks have evolved a microbial strategy, with the help of some probiotic bacteria, to maintain their health on the plateau.

### 4.1. Interaction between Yak Health and Yak Microbes

#### 4.1.1. Impact of Yak Gut Microbes on Yak Health

Yak gut microbes influence yak health [47], and the effects can be divided into three types: metabolism-dependent, immune, and neural activation pathways [48]. In metabolically dependent pathways, microbial communities affect intestinal epithelial cells through metabolites and regulate intestinal signaling pathways, such as butyrate produced by *Vibrio butyrate*, which alleviates symptoms of diabetes [49]. Short-chain fatty acids produced by Ruminococcus enhance host immunity and inhibit colonization by pathogenic bacteria by regulating intestinal pH [48,50,51]. In the immune pathway, to mitigate the effects of low temperature on yak gut health, cold stimulation increases the abundance of some *Akkermansia* spp., such as *Akkermansia muciniphila*, which are involved in maintaining the integrity of the host gut barrier. They also induce the differentiation of intestinal regulatory T cells, including T follicular helper cells, which play a crucial role in maintaining intestinal homeostasis by regulating immune responses and by mediating the balance of microorganisms in the gut [52,53,54]. In the neural activation pathway, probiotics stimulate the central nervous system through the enteric nervous system and vagus nerve; however, there are few studies on the neural activation pathway, and further studies are needed to clarify the mechanisms involved [55,56].

#### 4.1.2. Impact of Yak Health on Gut Microbiota

Yak health affects gut microbes. For example, diarrhea causes the dysregulation of gut microbes, resulting in the alterations in the relative abundances of some bacteria and fungi in the gut. This effect can be amplified by microbial networks, affecting more fungi and bacteria, thus increasing the adverse effects of diarrhea on yak gut health [57]. Gut microbes are also affected when yaks are under stress. For example, when yaks are stressed during transport, there is a reduction in the abundance of ruminal Prevotella, which affects the breakdown of carbohydrates and proteins [58]. With the stress at weaning in yak calves, the relative abundances of bacteria such as Firmicutes, Bacteroidetes, and 5-7N15 change in the feces [59]. In addition, environmental factors influence disease resistance in yaks. Wild yaks that live in environments with greater microbial diversity are more resistant to diseases such as diarrhea than domestic yaks [60]. The study of microbial associations with yak health may provide a reference for reducing the impact of the harsh highland environment.

### 4.2. Gut Probiotics in Yaks

*Bacillus subtilis* and Lactobacilli in the yak rumen inhibit the proliferation of pathogenic bacteria and improve the gut environment [61]. *Bacillus subtilis* is enriched in the ABC transporter protein-related gene pathway that reduces damage from toxic substrates [62] and protects the host [63]. Three strains of *Bacillus subtilis* (BS1, BS2, and BS3) and one of *Bacillus velezensis* (BV1) were isolated from the intestine of yaks. These four strains have growth-enhancing properties, can promote a healthy balance of intestinal flora, and can inhibit the growth of pathogenic bacteria [64]. *Bacillus pumilus* DX24 strain isolated from yak feces has growth-promoting properties and enhances antioxidant capacity by increasing superoxide dismutase (SOD) concentration in blood, and immune responses by increasing lysozyme (LZM) and alkaline phosphatase (AKP) activities [65].

Lactic acid bacteria can improve intestinal microbial structure and reduce the proportion of harmful bacteria in yaks [66]. For example, lactic acid bacteria can reduce the abundance of pathogenic bacteria Paenibacillus, Aerococcus, and Comamonas to reduce diarrhea caused by *E. coli* [67]. *Lactobacillus johnsonii* (LY1), which has a strong antibacterial effect against *E. coli* and *S. enteritidis*, and *Leuconostoc pseudomesenteroides* (P1), which has a strong antibacterial effect against *S. aureus*, were isolated from yak feces. The presence of LY1 and P1 can reduce the occurrence of diarrhea associated with bacterial diseases in yaks [68]. Five LAB strains *Leuconostoc mesenteroides*, *Lactobacillus plantarum*, *Enterococcus hirae*, *Lacticaseibacillus camelliae*, and *Lactobacillus mucosae*, isolated from yak vagina, displayed growth resistance, aggregation capacity, and potent antimicrobial activity against *E. coli*, *S. aureus* and *S. enteritidis* [69]. Therefore, lactic acid bacteria are important for the health of yaks, especially, to reduce diarrhea [66].

## 5. Relationship between Microorganisms and Quality of Yak Products

Like all ruminants, the yak rumen is a fermentation organ that converts substances that humans cannot, such as plant fibers, into valuable animal products such as meat, milk, and wool through microorganisms in the rumen. Yak milk is characterized by a high fat content, while yak meat is high in protein, low in fat, and has a unique flavor, which is closely linked to the specific microbial community in the yak rumen [70].

### 5.1. Microbiological Mechanisms of High Milk Fat Content in Yak Milk

Milk yield and milk composition in yaks are highly correlated with the abundance of different bacteria in the rumen. *Clostridium* spp. and *Ruminalococcus* spp. play essential roles in the fermentation of cellulose- and starch-rich diets, and in the production of acetic and butyric acids, which are precursors for the synthesis of milk lipids [71,72]. The microbial diversity of yaks is richer than in cattle, and the enrichment of their rumen bacterial genes in the bacterial chemotaxis pathway suggests that yak microorganisms, aided by flagella, move to and gather in areas with high nutrient concentration, which is conducive to improving the conversion of nutrients [73]. In addition, high bacterial diversity enhances milk fat synthesis [74,75]. Yaks from different geographical regions have specific gut microbial communities, which influence energy acquisition and, thus, milk composition through their enzymatic activities [76].

### 5.2. Microbiological Mechanisms of High Meat Quality in Yaks

The rumen microbial communities play a crucial role in energy storage and fat deposition in animals; in particular, Firmicutes and Bacteroidetes affect intramuscular fat deposition, which is strongly correlated with muscle fatty acid composition. The ratio of Firmicutes and Bacteroidetes in the rumen of yaks is different from that of cattle, which results in a low fat content of in yak meat [77,78,79]. Some bacteria in the rumen of yaks, such as the phylum Tenericutes, affect the content of muscle fat in longus muscle. The abundance of Tenericutes gradually increases with an increase in age, and Tenericutes is correlated negatively with fat content in longus muscle [80,81], which can explain, at least in part, the low fat content of yak meat. The relationship between microbiota and yak meat quality is not well established, and more research is needed to determine how microorganisms affect the taste and quality of yak meat.

## 6. The Role of Microorganisms in Yak-Environment Interactions

The Qinghai-Tibetan plateau is a fragile ecosystem and yaks are important in maintaining the ecological balance. The yak rumen is a conversion station for minerals and nutrients from soil, pasture, and feces, and a site for microbial exchange. The microorganisms in the rumen affect the soil and pasture through feces, and the environment by participating in metabolic processes, such as methane emission. The yak’s microorganisms can spread horizontally throughout the plateau ecosystem and affect other animals.

### 6.1. Microbiological Mechanisms of Low Methane Emissions in Yaks

Yaks emit less methane than cattle due to their microbial metabolic strategies. Yaks follow the ‘low carbon strategy’ to improve energy efficiency and the microbial strategy to compete for hydrogen with the methane emission pathway.

#### 6.1.1. Low Carbon Strategies

The “low carbon strategy” refers to yaks producing more volatile fatty acids, and emitting less methane than other ruminants [6]. Genome sequencing results revealed that 36 genes related to volatile fatty acid transport and uptake were up-regulated in the rumen epithelium of high-altitude ruminants compared to low-altitude ruminants. The rumen microbiota of yaks were enriched in volatile fatty acid-producing enzymes along the prokaryotic carbon sequestration pathway, which promoted efficient synthesis of volatile fatty acids [45]. Prevotella is involved in protein and hemicellulose degradation and the production of volatile fatty acids [82]. When yaks consume roughage, the abundances of some cellulose-degrading bacteria, such as *Coccidioides tumefaciens*, increase in the yak rumen, which secrete enzymes to break down fiber into acetic acid, thereby increasing the production of volatile fatty acids [83]. The high volatile fatty acid production reduces methane production by competing with the methanogenic pathway for hydrogen [45].

#### 6.1.2. Microbial Strategies Competing with Methane Emission Pathways for Hydrogen

The difference in methane emission between yaks and cattle is due mainly to their different enrichment relationships between hydrogen-producing microorganisms, hydrogen-consuming microorganisms and methanogenic hydrogen-nutrient microorganisms [84]. *Selenomonas*, enriched in the rumen with low methane emission, is involved mainly in the furanate and nitrate reduction metabolic pathways, which are hydrogen-consuming, and, therefore, can effectively compete with methanogenic bacteria for H2 [85]. Methane-producing hydrogenotrophic microorganisms such as *Quinella* spp. are more abundant in the rumen of yaks than in cattle [79]. The primary fermentation products of *Quinella* spp. are lactic and propionic acids, which produce little to no hydrogen during fermentation, and a reduction in rumen hydrogen results in less methane emission [86].

In ruminants with high methane emissions, protozoan enrichment in hydrogenase metabolic pathways were detected, which increases rumen hydrogen production, while no similar phenomenon was detected in ruminants with low methane emissions [87,88], thus, it can be speculated that the abundance of hydrogen-producing protozoa in the yak rumen is low. In fermentation studies on rumen fluid from yak and cattle, hydrogen utilization was greater in cattle than yak, and a high utilization of hydrogen leads to an increase in methane production [89]. In addition, the diversity of methanogenic bacteria (methanogens) is greater in yaks than in cattle, and the greater the diversity of methanogens, the lower the methane production [3]. For example, in an in vitro gas production assay, rumen fluid with a high diversity of sequences related to thermoplasmatales-affiliated linage C (TALC) methanogens produced less methane [90]. These characteristics explain why yaks have relatively low methane emissions and energy needs (Figure 2).

### 6.2. Interactions between Yak Gut Microbes and Soil Microbes

Soil has a vast reservoir of microbial diversity, and is considered the most genetically diverse ecosystem on Earth [91]. Microbes in soil influence microbes in yak rumen and vice versa; however, based on SourceTracker modeling, yak gut microbes have a greater influence on soil and grass microbes than soil and grass microbes have on yak gut microbes [91].

#### 6.2.1. Effects of Yak Gut Microbes on Soil

The diversity and abundance of soil bacteria and fungi on the Qinghai-Tibetan plateau are reduced by yak grazing and yak feces [92]. The abundance of beneficial microorganisms in fecal-contaminated soil declines, leading to adverse effects on soil nutrient cycling, soil pollution remediation, and rhizosphere stability, and a decline in soil health [93]. Concomitantly, the abundance of pathogens in grazed soils increases, lessening soil function and system resilience [92]. Compared to grazed soil, non-grazed soil has greater abundances of Acidobacteria and Latescibacteria, which play beneficial roles in soil restoration, plant growth, and nutrient cycling [94]. Yak grazing reduces soil lignin content and the abundance of trophic bacterial phyla responsible for decomposing lignin and maintaining ecosystem material cycling [95]. However, rational grazing can have a positive effect on grassland ecosystems, as the dung increases unstable carbon in the soil, which enables Ascomycetes to absorb organic matter and develop faster [96]. Ascomycetes are correlated positively with the diversity of grassland vegetation, so it is reasoned that rational grazing increases the diversity of pasture grasses [97].

#### 6.2.2. Effects of Soil Micro-Organisms on Yaks

The composition and abundance of soil micro-organisms can affect yaks. For example, an increased abundance of pathogens in the soil as a result of grazing can affect yak health. High abundance of *Desulfovibrio* can cause intestinal diseases [98], while genera such as *Romboutsia*, *Butyricimonas*, and *Parabacteroides* are increasing in abundance and pose a threat to host health [99,100]. The catabolism of soil microbial communities plays an essential role in trace elements and energy flow between plants. Soil microbes influence plants consumed by yaks and influence yaks, which, in turn, alter the composition of yak fecal microbes that influence soil microbes. In all fecal-contaminated soils, Ascomycetes is the dominant phylum, with γ-Ascomycetes containing a variety of pathogenic bacteria such as Salmonella, Yersinia, Vibrio, and *Escherichia* [101] that cause diarrhea [102].

### 6.3. Effects of Yak Microbes on People and Plateau Pika

#### 6.3.1. Effects of Yak Microbes on People Inhabiting the Qinghai-Tibetan Plateau

The yak is the major milk producer on the Qinghai-Tibetan plateau [103], and the milk and milk products are the primary foods consumed by the people [104], providing most of their vitamin and nutritional requirements [2]. As a result, the diet of highland people historically lacked fruits and vegetables, but the people did not suffer from vitamin and mineral deficiencies [105]. *Lactobacillus delbrueckii* subspecies bulgaricus F17 isolated from yak milk has a high free radical scavenging capacity and its consumption decelerates human oxidative stress [106]. The *Lactobacillus rhamnosus* CY12 strain from yak milk has a high survival rate in bile salts, acidic conditions, and gastrointestinal juices, and high antimicrobial activity and adhesion potential [107]. Yak yogurt is rich in microbial communities, particularly in beneficial lactic acid bacteria such as *Lactobacillus plantarum* Lp3 [108], which reduces the rate of cholesterol degradation [109,110]. Lactic acid bacteria with high antioxidant capacity in naturally fermented dairy products maintain their reductive stability by regulating the host’s antioxidant system, thus, reducing oxidative damage in the people residing on the plateau [111]. Yak yogurt has a lower lactose content than cow yogurt, which is beneficial for people with lactose intolerance [110].

#### 6.3.2. Effects of Yak Microbes on Plateau Pika

In the shared habitat of the Qinghai-Tibetan plateau, fecal microbes are crucial in the development of mechanisms that allow different species to coexist. For example, a small mammal known as the plateau pika (*Ochotona curzoniae*) is common on the Qinghai-Tibetan Plateau, feeding primarily on herbaceous plants [112]. Plateau pikas coexist with yaks, and consume yak dung in winter to compensate for food shortages [113], while yaks also graze near ground level. The vegetation in their shared habitat undergoes structural changes in response to animal activities, providing a more diverse food source for yaks and plateau pikas [114]. Multiple factors create mechanisms for microbes to establish mutually beneficial coexistence between yaks and plateau pikas [115]. For plateau pikas, yak transmission of Firmicutes, Bacteroidetes, Verrucomicrobia, and Proteobacteria enhance pika enrichment in prebiotic and immune-related pathways, and for yaks, horizontal transmission of bacteria by plateau pikas enhances pathways associated with hepatoprotection, exogenous biodegradation, and detoxification [115].

## 7. Conclusions

Yaks have co-evolved with microbes to withstand the extreme pressures of the harsh Qinghai-Tibetan plateau environment. These microbes are closely linked to the yak’s digestion, metabolism, and immune responses. Knowledge on the characteristics and functions of yak gut microbes is not complete. Future research should establish a yak microbiome network to examine the horizontal transmission pathways of microbes. In addition, by leveraging metagenomic sequencing technologies, mechanisms by which yaks adapt to extreme environments could be better understood. This could provide guidance in yak husbandry and understanding the adaptability of ruminants in extreme conditions.

## Figures and Tables

**Figure 1 microorganisms-12-01122-f001:**
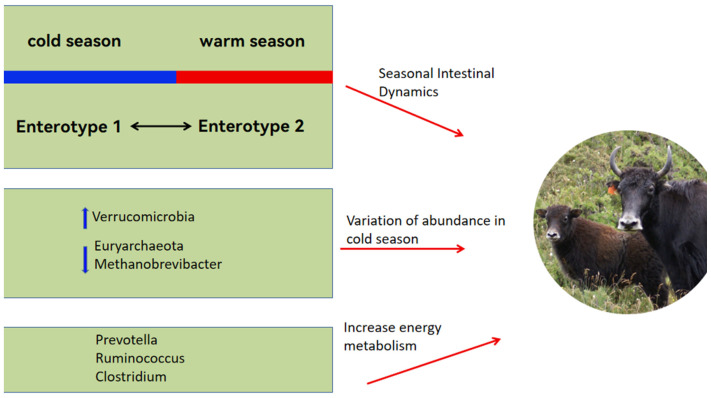
Yaks adapt to seasonal fluctuations in forage resources by altering the seasonal gut microbial compositions between the warm and cold seasons, thus enabling yaks to overcome scarce forage resources.

**Figure 2 microorganisms-12-01122-f002:**
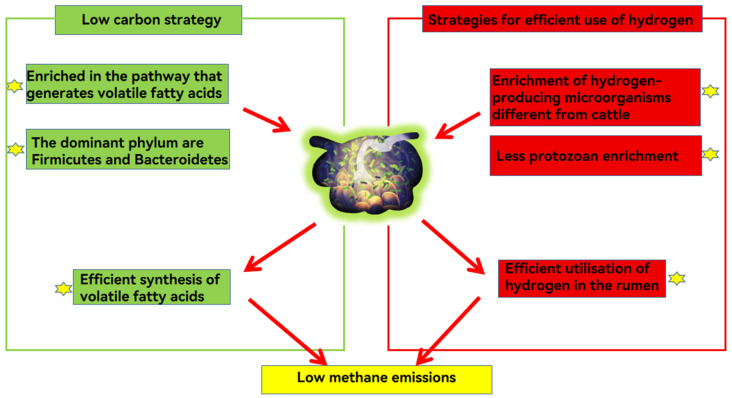
Yaks cope with the harsh environment of the Tibetan Plateau by improving energy efficiency through two strategies: efficient synthesis of volatile fatty acids and utilization of hydrogen by rumen microbes and reduction of rumen hydrogen production and methane emission.

**Table 1 microorganisms-12-01122-t001:** Dominant ruminal bacteria at the family level in yaks.

Family	Main Functions
Ruminococcaceae	Degrade fiber and proteins
Succinivibrionaceae	Degrade starch and fiber
Lachnospiraceae	Promote growth of fiber-degrading bacteria
Rikenellaceae	Degrade fiber
Bacteroidaceae	Degrade starch and fiber and improve fiber digestibility.
Prevotellaceae	One of the major glycolytic flora of the rumen, known for its protein binding capacity and digestion of a wide range of carbohydrate substrates
Christensenellaceae	Quickly respond to changes in feed components and participate in protein catabolism

## Data Availability

No new data were created or analyzed in this study. Data sharing is not applicable to this article.

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
