# Peer review of "Yaks Are Dependent on Gut Microbiota for Survival in the Environment of the Qinghai Tibet Plateau"

_microorganisms, 2024, doi:10.3390/microorganisms12061122_

Round 1

Reviewer 1 Report (Previous Reviewer 1)

Comments and Suggestions for Authors

only one thing: line 80 may be written microbiota

Author Response

Reviewer 2 Report (Previous Reviewer 2)

Comments and Suggestions for Authors

The article provides a comprehensive examination of the intricate relationship between yaks and microorganisms, establishing a robust platform for further scholarly investigation in this domain. It effectively underscores the critical significance of this subject matter, particularly in elucidating the pivotal role of microorganisms in influencing the adaptability of yaks, especially within the challenging environment of the Tibetan Plateau. By delineating the diverse mechanisms through which microorganisms impact yak physiology and metabolism, the article furnishes invaluable insights into the symbiotic alliance between yaks and their microbiota. To enhance the article, the integration of more quantitative data is recommended. Additionally, incorporating a dedicated section outlining potential avenues for future research is recommended.

Comments on the Quality of English Language

Minor editing of English language required

Author Response

Reviewer 3 Report (New Reviewer)

Comments and Suggestions for Authors

Dear authors, congratulations on preparing such an interesting manuscript.

In my opinion, the prepared one contains all the necessary elements to publish content.

Author Response

This manuscript is a resubmission of an earlier submission. The following is a list of the peer review reports and author responses from that submission.

Round 1

Reviewer 1 Report

Comments and Suggestions for Authors

Abstract Line 15 Is it feasible to establish a clear and robust objective?

Introduction Line 28 Is it appropriate to include a concise overview of the animal's typical diet in the introduction?

Line 62 The fungal structure undergoes continuous changes.

Line 76 Justified instead of warranted

Lune 80 … with better-quality pasture (such as) please name the main plants grazed by the yaks.

Line 104 source???

Line 116 Encoded typically refers to the process of converting information or data into a particular format for transmission, storage, or interpretation. Are you trying to say that?? Please give some brief explanation for better understanding.

Line 150 Verrucomicrobia is a phylum of bacteria that are characterized by their distinct cellular morphology and genetic makeup. These microorganisms typically have a unique outer layer and can be found in various environments, including soil, water, and the gastrointestinal tracts of animals. They play roles in nutrient cycling and ecological processes, and their study contributes to our understanding of microbial diversity and ecosystem dynamics. Please give an explanation like this…

Line 167 source of figure 1

Line 188 Please modify as this: T cells, including T follicular helper cells, play a crucial role in maintaining intestinal homeostasis by regulating immune responses and supporting the balance of microorganisms in the gut.

Line 243. Move toward and aggregate in regions with elevated nutrient levels

Line 268 Who accomplishes this task and by what means?

Line 286 …Among hydrogen-producing microorganisms and hydrogen-consuming organisms.

Line 306 again, source of figure 2

Line 332 microorganisms

Line 360 what do you mean by plateau pikas, please give some definition. Like this: Plateau pikas are small mammals belonging to the genus Ochotona, native to the Tibetan Plateau and surrounding regions.

Reviewer 2 Report

Comments and Suggestions for Authors

The presentation of information in the article is convoluted, making it difficult to grasp key concepts thoroughly. To enhance clarity and scientific rigor, it is essential to restructure the content in a more coherent and logical manner. Incorporating concrete examples to substantiate specific claims and elucidate complex microbiological phenomena would greatly improve understanding of the article. Furthermore, the author must enrich the analysis by comparing data from other ruminant species, which would significantly deepen understanding of the unique microbiological mechanisms observed in yaks and further develop  understanding of the distinctive yak microbiota and specific microbiological processes characterizing yaks. This comparative approach can help identify key factors facilitating adaptation of this animal in its environment.

The author should provide more detailed explanations of the discussed microbiological mechanisms, especially those specific to this species. Offering comprehensive explanations is crucial for clarifying nuances and facilitating deeper understanding.

Additionally, it is essential for the author to include quantitative data in the article to support the presented claims and strengthen the scientific basis of the findings.

Comments on the Quality of English Language

 Minor editing of English language required